# Biomarkers Associated with Drugs for the Treatment of Lupus Nephritis

**DOI:** 10.3390/biom13111601

**Published:** 2023-10-31

**Authors:** Huiyu Nie, Siyuan Chang, Yuanyuan Li, Fen Li

**Affiliations:** 1Department of Rheumatology and Immunology, The Second Xiangya Hospital of Central South University, Changsha 410011, China; 2Clinical Medical Research Center for Systemic Autoimmune Diseases in Hunan Province, Changsha 410011, China

**Keywords:** biomarkers, lupus nephritis, treatment, belimumab, telitacicept

## Abstract

The constant updating of lupus drug treatment guidelines has led to a question. How can the efficacy of treatment be more effectively monitored? Systemic lupus erythematosus (SLE) is a complex autoimmune disease that often presents clinically with multi-organ involvement, and approximately 30% of patients with SLE develop lupus nephritis (LN). Therefore, it is important to better track disease progression and drug efficacy. Now, kidney biopsy is still the gold standard for diagnosing and guiding the treatment of LN, but it is invasive and expensive. If simple, non-invasive and effective biomarkers can be found, drug intervention and prognosis can be better monitored and targeted. In this review, we focus on LN and explore biomarkers related to LN therapeutics, providing clinicians with more possibilities to track the therapeutic effect of drugs, improve treatment options and assess patient outcomes.

## 1. Introduction

Systemic lupus erythematosus (SLE) is a multi-organic, multi-system involvement of autoimmune diseases. Lupus nephritis (LN) is one of the common clinical manifestations of SLE, and some patients present renal involvement as the first symptom, mainly manifested as hematuria, proteinuria, edema and hypertension [1]. It is particularly important to find reliable indicators to predict the risk of LN in SLE patients [2]. As a biochemical indicator, biomarkers can diagnose, monitor and predict prognosis of diseases, or effectively evaluate the efficacy, safety and adverse reactions of drugs [3]. In recent years, omics technology is an important branch of rapid development in the field of medical development, mainly including genomics, transcriptomics, proteomics, metabolomics and lipidomics. These various types of omics discover differentially expressed biomarkers through different technical means, so as to better apply them to clinical practice. At present, the classic clinical indicators are used for the diagnosing and monitoring of LN, which are mainly autoantibodies and complement. Most of the novel biomarkers are still in the stage of early diagnosis and disease surveillance, and no biomarker can predict LN disease progression and treatment response with high reliability and reproducibility.

Although the pathogenesis of LN is not well understood, it is related to genetics, environmental factors, immunity, neutrophil extracellular trap (NET) and other factors (Figure 1). Through single-cell transcriptomics techniques, a study revealed the complexity of the cell population in LN. The interactions and expression markers between different cells provide more avenues for treating LN [4]. The various pathological changes in the body and the therapeutic decisions vary greatly. By combining various novel and traditional techniques to find promising biomarkers, treatment can be better guided.

We conducted an electronic literature search from 1961 to 2022 in Web of Science, Embase and PubMed. We mainly used the following terms: lupus nephritis, biomarkers, treatment. We try to review biomarkers related to the conventional and biologic therapeutic drugs (in chronological order). These biomarkers are of various types and have different mechanisms of action. They may provide a deeper understanding of the disease in terms of its onset, progression, outcome and prognosis. 

## 2. Biomarkers of Conventional Drugs for the Treatment of LN 

LN treatment has a long history. The emergence of glucocorticoid (GC) was a landmark. However, GC therapy has a low remission rate and high side effects; a variety of immunosuppressive agents have emerged for the treatment of LN.

### 2.1. Biomarkers of GC Alone

In the 1960s, GC was the first drug used for LN treatment, especially the short-acting GC. In 1961, it was suggested that the symptoms of LN patients could be relieved by using high doses of prednisone [5]. However, in recent years, the treatment guidelines no longer advocate the use of high-dose GC alone, but the combination of GC and immunosuppressive induction therapy, so the research related to it is also relatively early. 

Podocytes, which are epithelial cells of renal vesicles with foot processes, mainly regulate the filtration function of the glomerulus. In 2000, Japanese scholars suggested that urinary podocytes could be used to indicate the disease activity of LN, and found that podocytes were only presented in the urine of patients with active LN compared to the control group [6]. In addition, when the team tested the substance in urine again after intravenous methylprednisolone (1.0 g/day) for 3 days and then prednisolone (1.0 mg/kg/day) for maintenance therapy in patients with active LN, the results showed that the urinary podocytes disappeared. However, we need to note the changes in urinary podocytes by GC do not fully represent that urinary podocytes are only affected by GC.

Urinary N-Acetyl-β-D-glucosaminidase (NAG) is mainly present in lysosomes in proximal tubular epithelial cells, and urinary NAG excretion is related to a variety of kidney diseases. In a study [7], the group of patients with LN or rheumatoid arthritis (RA) showed urinary NAG excretion, as compared to the healthy control group. In the LN group, NAG excretion was higher than in the RA group and healthy control. In a trial of Gluhovschi, C et al. [8], urine testing in patients with LN before treatment and on Days 7 and 30 after treatment with oral prednisone (1 mg/kg) showed a significant reduction in NAG, but the improvement occurred later than in proteinuria and glomerular filtration rate. The above trials suggested that the measurement of urine NAG may be a useful complement to routine biochemical analysis of urine, but whether it can be used as a biomarker to monitor the efficacy of GC needed to be further explored. 

Fatty acid esters of hydroxy fatty acids (FAHFAs) are a newly discovered class of biologically active lipid molecules. Abnormal lipid metabolism is one of the pathogenic mechanisms of LN. In a study using lipidomics techniques in an LN mice mode [9], the team observed that the total levels of FAHFAs were significantly reduced in mice treated with GC compared to those not treated with GC. However, the study also highlighted that the long-term use of GC could worsen abnormal lipid metabolism. Therefore, FAHFAs may serve as potential markers to determine when to discontinue GC therapy. 

### 2.2. Biomarkers of Hydroxychloroquine Alone

Hydroxychloroquine (HCQ) can exert therapeutic effects by regulating the proliferation and differentiation of lymphocytes and inhibiting the production of pathogenic cytokines. S100A8 and S100A9 are two calcium-binding proteins of the S100 family that can induce the development of renal immune responses by binding to interactive pattern-recognition receptors, such as toll-like receptors. A single-center retrospective study initiated by Wakiya, R. et al. [10], in which patients were in low disease activity at the time of enrollment, had no organ involvement and were treated with HCQ alone for at least 3 months, found that S100A8 and S100A9 were abnormally elevated at baseline, but reduced after 3 and 6 months of treatment with HCQ. Perhaps S100A8 and S100A9 can be used as biomarkers with the clinical value of HCQ for LN treatment and follow-up.

Proprotein convertase subtilisin/kexin type 9 (PCSK9) is an enzyme encoded by the PCSK9 gene that binds to low-density lipoprotein (LDL) receptors, resulting in an abnormal increase in LDL and affecting lipid metabolism. Fang et al. found that serum PCSK9 levels were elevated in SLE patients, particularly in those with atherosclerosis or LN, but significantly reduced in inactive SLE patients after three months of monotherapy with HCQ [11]. Although HCQ is a good drug for preventing the recurrence of LN, using it alone cannot completely suppress the condition of LN without recurrence, so HCQ cannot be used alone in clinical practice.

### 2.3. Biomarkers Related to Combined Treatment of LN

For LN patients, the long-term use of GC may lead to different kinds of side effects, such as obesity, osteoporosis and risk of infection. The combination therapy with GC and immunosuppressive agents has the following advantages. Firstly, it reduces the respective drug doses and decreases the incidence of adverse reactions. Secondly, it exploits the synergistic effects of drugs and enhances the therapeutic effect. Finally, it prevents or delays the emergence of drug resistance and reduces the risk of recurrence after stopping the drug. Therefore, we focus on the summary of biomarkers in combined therapies that are more closely related to the clinical practice of LN administration. We put emphasis on the biomarkers of combined therapies to LN administration in clinical practice. 

In the 1990s, immunosuppressive agents were becoming increasingly important in the treatment of LN. The 2019 Joint European League Against Rheumatism and European Renal Association-European Dialysis and Transplant Association (EULAR/ERA-EDTA) highlighted the importance of cyclophosphamide (CTX) and mycophenolate mofetil (MMF) [12]. There are few studies on biomarkers of the treatment of LN by immunosuppressive agents alone, only in CTX and MMF. There are no studies related to azathioprine (AZA), cyclosporine A (CsA), voclosporin and tacrolimus.

#### 2.3.1. Glucocorticoid and Cyclophosphamide

CTX is effective in both induction and maintenance treatment with moderate-to-severe LN patients, especially for patients with LN type III and IV (with or without V). The 2019 EULAR/ERA-EDTA recommended GC and CTX in combination [12].

C1q is involved in initiating the classical activation pathway. Serum C1q levels had been found to be a predictor of histopathological outcomes when patients were treated with prednisolone and CTX at first renal biopsy and re-renal biopsy after 6 months [13]. In this study, it found that when serum C1q levels were low, the results of renal biopsy histopathology in LN patients may show further progression or manifestations in the active phase, resulting in a poor response to treatment. Anti-C1q antibodies levels in serum increased before renal involvement [14,15]. Some researchers believed that continuous monitoring of anti-C1q antibodies levels could guide treatment [14]. However, some researchers believed that monitoring the level of anti-C1q antibodies was an important means of diagnosing LN, but it could not be used to monitor the course and treatment [16]. The reason for the controversy between the two views may be related to the length of observation in both studies.

In 2018, Guleria A et al. [17] conducted an analysis of serum metabolite levels in 18 patients with proliferative LN using nuclear magnetic resonance-based metabolomics. The study examined the effects of high-dose or low-dose CTX maintenance therapy in combination with prednisolone over a period of 6 months. Following the treatment, the level of lipid metabolites, predominantly LDL and very-low-density lipoproteins (VLDL), in the serum decreased significantly, while the level of acetate increased. Building upon this research, the same team conducted a subsequent study in 2020, employing nuclear magnetic resonance-based metabolomics technology to analyze the urinary excretion levels of citrate and acetate in 18 LN patients. The patients were treated either with the National Institutes of Health regimen or the Eurolupus regimen [18]. The results revealed notable changes in urine citrate levels after 6 months of treatment, compared to a healthy control group. Prior to treatment, the citrate levels were significantly reduced in comparison to the healthy control, but showed a significant increase after treatment. Although acetate levels also increased, the change was not statistically significant. It is worth noting that further research centers may be required to validate the reliability of these findings.

Hepatocyte growth factor (HGF) is a multifunctional factor and plays an important role in renal tubular epithelial cells [19]. Transforming growth factor β1 (TGFβ1) is a cytokine in the TGFβ family which can inhibit the function of a variety of immune cells. A study suggested that tubular injury could be assessed by the ratio of HGF to TGFβ1, and when HGF levels were elevated and TGFβ1 levels were reduced, the response of LN patients to low-dose GC and CTX treatment after 6 months would be better [20].

CTX is commonly used as a treatment for refractory LN and is thought to improve short-term prognosis. Asymmetric dimethylarginine (ADMA) is a hydrolysis product of arginine methylation modification. ADMA can inhibit nitric oxide production by blocking the activity of nitric oxide synthase, which may lead to reduced glomerular blood flow and affect kidney function [21]. A study pointed out that the levels of ADMA in plasma in proliferative LN patients with or without membranous LN were significantly increased. Patients in remission after CTX treatment had significantly lower levels of ADMA than those who did not respond. Thus, the levels of ADMA in plasma may be used as a non-invasive indicator to predict the effect of CTX therapy in LN patients [22].

#### 2.3.2. Glucocorticoid and Mycophenolate Mofetil

MMF is a 2-ethyl ester derivative of mycophenolic acid (MPA). When MMF enters the body and is absorbed, it is converted into MPA, which exerts an inhibitory effect on the immune response by specifically inhibiting the lymphocyte purine de novo synthesis pathway. Glycosphingolipids (GSL), a type of sphingolipid, plays an important role in cell adhesion, growth and differentiation, as well as intercellular signaling. It was confirmed that GSL levels were elevated in the urine of LN mice and patients [23]. Baseline levels of GSL in urine extracellular vesicles had been found to be a predictor of whether complete remission could be achieved in LN patients [24]. In this trial, LN patients in the MMF and placebo groups in the LUNAR and abatacept clinical trials, and LN patients treated with MMF at the MUCS Lupus Clinic, showed that LN patients who did not achieve complete response by treating with MMF had higher baseline levels of GSL than those who achieved complete remission.

#### 2.3.3. Glucocorticoid and Multiple Immunosuppressive Agents

In addition to CTX, other immunosuppressive agents are equally important. When LN patients treated with CTX develop serious side effects, other immunosuppressive agents may be used as alternative treatments.

Monocyte chemoattractant protein-1 (MCP-1) is mainly responsible for the chemotaxis of monocyte-macrophages and dendritic cells. Interleukin-1 (IL-1) can induce glomerular cells to produce MCP-1, so it can indirectly reflect the expression of MCP-1 in the kidney [25]. A number of studies suggested that MCP-1 could be used as a biomarker reflecting disease activity and prognosis in LN patients [26,27,28]. However, relying on only the detection index, its sensitivity and specificity could not be fully guaranteed, so using the combination of MCP-1 and the tumor necrosis factor-like weak inducer of apoptosis (TWEAK) as a combination biomarker has become a research point worthy of attention [28,29,30]. TWEAK, as a member of the tumor necrosis factor superfamily, has the effect of killing tumors and participating in the inflammatory response, which can lead to glomerular and tubular damage [31]. In South Africa, GC and (CTX or MMF) was used with 20 active LN patients [32]. It found that the levels of MCP-1 and TWEAK in urine decreased significantly in LN patients who achieved partial or complete remission, and the predictive efficiency of the combination of the two (MCP-1 and TWEAK) was better than the single index, which is a potential clinical model forecasting LN patients’ remission that needs to be explored in depth. In addition, a study investigated the role of soluble tumor necrosis factor receptor 2 (sTNFR2) in LN patients [33]. The study found that before treatment, the baseline levels of serum sTNFR2 were higher in 64 LN patients compared to the control group. After treatment with GC and [CTX or MMF or AZA or Rituximab (RTX)], the levels of sTNFR2 decreased in the group of patients with proliferative LN. However, in the group of patients with membranous LN, the levels of sTNFR2 only decreased in some patients who responded positively to the treatment. Therefore, whether sTNFR2 can be used as a predictor of LN treatment outcome needs to be further evaluated.

## 3. Biomarkers of Biological Drugs for the Treatment of LN

In comparison to conventional medications, biological drugs possess several advantages, including a heightened specificity, enhanced responsiveness and diminished incidence of adverse effects. In this chapter, we mainly introduce the drugs in the two major aspects of immune cells and inflammatory factors, and briefly introduce some new small-molecule inhibitors. However, it should be noted that the price of biological drugs is expensive. Thus, biomarkers related to the treatment of LN patients with biological drugs have not been studied by histologic techniques. This may be a worthwhile direction for future research.

### 3.1. B Cells Targeting Drugs

B cells play an important role in the pathogenesis of LN; when abnormal B cells differentiate and mature, they release a large number of auto-antibodies and attack the kidney tissues in vivo, thus causing kidney damage. There are many B cell targets, including B-cell-activating factor (BAFF), A proliferating inducing ligand (APRIL), CD20 and CD19. Related drugs include belimumab (BLM), atacicept, telitacicept, ALPN-303 and RTX. Although the mechanism of action is somewhat different, they all inhibit the function of B cells by binding to the corresponding biologically active substances.

#### 3.1.1. Targeted Inhibitors of BAFF

BAFF is a cytokine necessary for B-cell maturation and survival, which exerts immune effects by binding to the B-cell-activating factor receptor (BAFF-R) on the surface of B cells [34]. BLM, a fully human recombinant IgG1-λ monoclonal antibody, works by interacting with BAFF, preventing the binding of BAFF to BAFF-R, thereby reducing the number of naive and transitional B cells and improving B cell function. In addition, there are novel inhibitors that act on BAFF targets, such as tabalumab and blisibimod. However, the phase III trial for tabalumab did not achieve the desired effect and provide a significant clinical benefit compared with placebo [35]. The mechanism of action of blisibimod is similar to tabalumab, and its phase III trial did not achieve an SLE response index-6, but it was well tolerated and safe [36].

A phase III trial for BLM enrolled 562 patients [37]. The baseline demographics, disease activity and biomarkers of patients with or without seizures at Weeks 24 and 52 of treatment were assessed. The results showed that the anti-dsDNA antibody ≥ 200 IU/mL was an independent predictor of episodes in patients with moderate-to-severe SLE at Week 52, while a low C3 level was an independent predictor of severe lupus episodes at Week 52. Another clinical trial of BLM included 21 patients [38]. Clinical evaluation and laboratory testing at baseline, 6 months and 12 months showed that serum levels of BAFF at baseline were the best predictor of response to BLM. BAFF is a specific biomarker that may help to identify and personalize treatment in patients who are more effective in treatment. In addition, a BLM phase III trial enrolled in 142 patients [39]. It aimed to evaluate its efficacy and safety in the East Asian BLISS-LN subgroup with a placebo or BLM and standard of care (oral GC and CTX induction followed by AZA maintenance, or MMF induction and maintenance therapy). The results showed that the anti-dsDNA antibody and anti-C1q antibody levels in the BLM group decreased more than in the placebo group but the C3 and C4 levels had a more pronounced rise than in the placebo group. 

#### 3.1.2. Targeted Inhibitors of BAFF/APRIL

APRIL, similar to BAFF, is also a cytokine that plays an important role in the activation and survival of B cells. Atacicept is the earliest biologic to target BAFF and APRIL factors, and its emergence has opened a new era of dual-target therapy, but because it did not reach the efficacy endpoint in lupus patients, had large side effects and had the possibility of increasing the infection rate [40,41]. Therefore, it was not yet available. However, it did not affect the boom of dual-target drug research. The drug telitacicept led by RemeGen Co., Ltd. (Yantai, China), was a new generation of dual-target biologics targeting B cells, which had been conditionally approved for marketing in China due to its safety, tolerability and effectiveness in early clinical trials [42]. Therefore, its in-depth study is worthy of attention and can help improve the level of drug efficacy in the real world.

Telitacicept has been on the market for a short time and is only available in China. There has no in-depth study of biomarkers related to telitacicept with LN. A telitacicept phase I trial enrolled 23 patients [43]. By evaluating the pharmacokinetic analysis of the drug in patients with lupus and RA, it pointed out that the establishment of the mechanistic target-mediated drug disposition model was considered to be the best choice for studying the changes in BAFF as the evaluation of efficacy, and the IgG level was also considered to be an important covariate affecting the baseline levels of BAFF. In addition, ALPN-303, a new pilot drug targeting the BAFF/APRIL dual target, appears to be more effective than telitacicept, and its preclinical study noted that serum levels of IgM, IgG and IgA antibodies showed a significant and sustained reduction after the administration of ALPN-303 in spontaneous LN mice and cynomolgus monkeys [44].

#### 3.1.3. Targeted Inhibitors of CD20

CD20 is a transmembrane phosphoprotein located on the surface of B lymphocytes and regulates the proliferation and differentiation of B cells. In the treatment of LN, there are many anti-CD20 monoclonal antibodies. RTX is the earliest anti-CD20 monoclonal antibody for the treatment of LN.

As early as 2002, observational studies showed that RTX was effective with lupus patients but the clinical trials EXPLORER [45] and LUNAR [46] did not meet the final clinical endpoint. Nevertheless, RTX treatment of LN still had a significant clinical efficacy and good safety. It was a promising treatment [47]. Compared to RTX, obinutuzumab, as a third-generation anti-CD20 therapeutic monoclonal antibody, enhances the depletion mechanism of B cells and has a better affinity for receptors, thereby improving the therapeutic effect. The obinutuzumab phase III trial enrolled 125 patients [48]. The result showed that compared with standard treatment alone, obinutuzumab and standard of care could improve renal function, and had the characteristics of good tolerability and high safety. Obinutuzumab is a new drug worthy of in-depth exploration.

Some researchers compared and analyzed serum anti-C1q antibody and antineutrophil cytoplasmic antibodies (ANCA) antibody levels after RTX alone or (RTX with CTX) treatment of LN patients [49]. The results showed that RTX combined with CTX was better than CTX in the treatment of LN, significantly reducing the levels of serum anti-C1q antibodies and ANCA, and significantly improving the prognosis of LN, indicating that the reduced levels of serum anti-C1q antibodies and ANCA with RTX were directly related to the improvement of disease prognosis, and also meant that they may be used as biomarkers to judge disease prognosis and drug efficacy. Tineke Kraaij et al. investigated a newly developed method for measuring C4d [50]. The trial evaluated the performance of serum C4d, C4 and the C4d to C4 ratio in 15 LN patients and then showed that the relationship between C4d and C4 levels was a valuable biomarker for longitudinal monitoring of treatment.

There are many types of urine proteins screened for LN, but different proteins have complex and diverse functions. There are many factors that affect the results of experiments, so there is no urine protein or proteome that has been applied to the clinical practice of LN. A research team believed that a urine proteome model consisting of lipocalin-like prostaglandin D synthase (LPGDS), transferrin, alpha-1-acid glycoprotein (AGP-1), ceruloplasmin, MCP-1 and soluble vascular cell adhesion molecule-1 (sVCAM-1) may be used as biomarkers for active LN and predict the efficacy of RTX [51,52].

#### 3.1.4. Targeted Inhibitors of CD19

CD19 is a surface protein located in B lymphocytes and follicular dendritic cells, which is mainly expressed in acute or chronic lymphocytic leukemia and B-cell lymphoma. In recent years, CD19 chimeric antigen receptor T cell (CAR-T) therapy has come into the limelight, mainly for the treatment of cancer. In 2021, researchers in Germany induced rapid remission in patients with refractory SLE with active LN by using CAR-T therapy. The levels of urine protein and anti-dsDNA antibodies were significantly reduced, and the levels of C3 and C4 also returned to normal, which may become a new method for treating patients with refractory LN [53].

### 3.2. T-B Cells Targeting Drugs

CD40 and CD40L are co-stimulatory molecules expressed on the surface of B cells and associated with the second signaling of B-cell activation. When T cells interact with B cells, CD40L on the surface of T cells binds to CD40 molecules on the surface of B cells to form a complex. This complex promotes adequate activation of B cells, thereby enhancing their ability to respond to antigens. Currently, the drugs under study are BI655064, CFZ533 and dapirolizumab.

#### 3.2.1. Targeted Inhibitors of CD40

There is expression of the co-stimulatory molecule CD40 on the surface of many kinds of immune and non-immune cells, which induces the activation of B cells and the production of a large number of autoantibodies by binding to CD40L on the surface of CD4+ helper T cells. How to effectively block CD40-CD40L mediated signaling pathway has become a hot spot in treatment. BI 655064 is a humanized IgG1 CD40 monoclonal antibody that has shown good levels of safety and tolerability in phase I trials [54]. Currently, a double-blind, randomized, placebo-controlled trial evaluating the effect of BI655064 subcutaneous injection on renal response one year after treatment in patients with active LN (NCT02770170) and an exploratory maintenance trial evaluating the effect of BI655064 on patients with LN (NCT03385564) have both been completed, perhaps as a potential drug for the treatment of LN, but larger multicenter trials are needed to validate. CFZ533 is another monoclonal antibody against CD40, and a preliminary clinical trial initiated by Novartis Pharmaceuticals on the safety, pharmacokinetics and efficacy of CFZ533 in LN patients has completed (NCT03610516). The results can be expected.

#### 3.2.2. Targeted Inhibitors of CD40L

The interaction of the CD40-CD40 ligand (CD40L) can active B cells. In the phase I clinical trial of the CD40L blockade dapirolizumab pegol (DZP) in SLE patients, researchers found that there were more apparent changes in responders’ peripheral blood than non-responders and the placebo group in mRNA transcripts, including Ig-associated genes, type I IFN-response genes and B cells’ surface protein genes [55]. Although the original intention of the transcript test might be verifying the mechanism of drug, we can still speculate that this result suggests the potential of these genes to become the biomarker of DZP, as mRNA is a common type of biomarker and peripheral blood is also one of the accessible specimens.

### 3.3. Targeted Inflammatory Factors

There are many inflammatory factors, and those closely related to the treatment of LN mainly include interferon (IFN) and interleukin (IL). Inflammatory factor inhibitors reduce the inflammatory response by reducing the number of these inflammatory factors. In addition, Janus kinase (JAK) is a non-receptor tyrosine kinase that acts on the inflammatory signaling pathway, and JAK inhibitors block the production of inflammatory factors by inhibiting inflammatory signaling.

#### 3.3.1. Interferon Inhibitors

IFN mainly includes three types (IFN-α, IFN-β, IFN-γ), which are further divided into type I IFN (IFN-α, IFN-β) and type II IFN (IFN-γ). Type I IFN is mainly responsible for antiviral responses, and type II IFN is mainly responsible for immune regulation. Omalizumab, a drug targeted to IgE, can reduce the production of type I IFN by inhibiting plasma cell-like dendritic cells and basophil activation; the results of its phase I trial showed the drug was well tolerated with lupus patients and could improve disease activity, so larger clinical trials were needed to prove its efficacy and safety [56]. IFN-α was an important pathogenic factor in the occurrence and development of LN. Although it was elevated in the serum of patients, it could not be used as a biomarker of disease activity [57]. Current IFN-α monoclonal antibodies include sifalimumab, rontalizumab, anifrolumab, IFN-α-kinoid and ASG-009. Among them, sifalimumab had completed phase I and II trials [58,59]. It has achieved good results and is a promising and research-worthy drug. Lupus low disease activity (LLDAS) is an important direction for LN treatment. Anifrolumab achieved its goal of a shorter duration of LLDAS and longer duration of LLDAS for the first time [60]. In 2021, the Food and Drug Administration (FDA) approved AstraZeneca’s developed Anifrolumab [61]. In addition, a study showed that the expression of the renal long-chain non-coding RNA RP11-2B6.2 promoted the progression of the type I IFN signaling pathway in LN patients, which may be a new therapeutic target [62]. Emapalumab is a monoclonal antibody against IFN-γ, which is mainly used for the treatment of primary hemophagocytic lymphohistiocytosis, and its efficacy against LN remains to be explored. Furthermore, a study believed that IFN-λ was a potential action factor of LN [63]. 

#### 3.3.2. Interleukin Inhibitors

IL is produced by immune cells such as lymphocytes and macrophages and is involved in the development of disease activity and inflammatory responses in patients with LN. 

IL-2, as the first molecular cloned cytokine, has a very important significance in the development and homeostasis of regulatory T cells, and it could treat LN by promoting the function of Treg [64]. More importantly, some studies believed that low-dose IL-2 (aldesleukin) could selectively induce the increase in Treg and better exert immune effects. Patients better tolerated this and it is a treatment option worth considering [65,66].

IL-6 is a multifunctional pro-inflammatory cytokine derived mainly from infiltrative inflammatory monocytes, macrophages and mesangial cells [67]. Baseline serum IL-6 levels were used as biomarkers for the early detection of disease activity in SLE patients and predictors of LN response and remission [68]. IL-6 monoclonal antibodies mainly include tocilizumab, sarilumab, clazakizumab, sirukumab, siltuximab and olokizumab. There are a few clinical studies of specific treatment of LN targeting IL-6. In a controlled study of patients with active LN treated by sirukumab, investigators found no trend observed for differences in serum IL-6 levels between responding and non-responding patients during the initiation of surveillance or at Week 24, although urine IL-6 levels differed significantly [69]. Therefore, the detection of serum IL-6 levels may not truly reflect the efficacy of IL-6 inhibitors, and indirectly reflect IL-6 levels, or urine IL-6 levels may be more predictive of the effectiveness and prognosis of treatment response in patients with LN, which requires more cohort studies to verify. 

Another IL, IL-17, is also a kind of proinflammatory cytokine which activates T cells and cytokine production to play a role in organ damage of LN. On the one hand, according to Abdel Galil, S.M et al. [68], IL-17 itself in serum could act as a biomarker of disease activity based on the comparison between active versus inactive groups, as well as of remission based on the comparison between active groups that were treated with a CTX infusion plus oral AZA and GC and remission groups that were treated with oral AZA, HCQ or GC. However, it is regrettable that there was no vertical observation result to be reported in this article which can apply IL-17 to be the biomarker of a specific treatment. On the other hand, several IL-17 monoclonal antibodies, such as ixekizumab, brodalumab and secukinumab, have completed their preclinical studies and are about to enter or have entered their clinical research phase in LN patients (NCT04181762 and NCT05232864), and studies to explore their biomarkers may follow closely.

#### 3.3.3. Janus Kinase Inhibitors

The JAK-signal transducer and activator of transcription (STAT) pathway is involved in many important biological processes such as cell proliferation, differentiation, apoptosis and immune regulation. There are four members of the JAK family (JAK1, JAK2, JAK3, TYK2), so there are many types of JAK inhibitors. But they mainly focused on the treatment of RA. For the treatment of LN, the main drugs include tofacitinib, baricitinib, upadacitinib, filgotinib and deucravacitinib. There are no drugs approved for LN. A study showed that TYK2 was not affected when regulating cytokines such as IL-23, IL-12 and type I IFN [70]. Therefore, by inhibiting TYK2, it was expected to reduce the side effects of inhibiting other members of the JAK family. BMS-986165 was an inhibitor that selectively binds to the pseudokinase domain of TYK2, which could effectively reduce the level of anti-dsDNA antibody titers in serum, as well as the infiltration of monocytes and IgG deposition in the kidneys of LN mice [71].

#### 3.3.4. Other Biological Small-Molecule Inhibitors

In recent years, several small molecules have been attempted for the treatment of SLE and LN, such as spleen tyrosine kinase (SYK) inhibitors (SYKi), Bruton tyrosine kinase (BTK) inhibitors (BTKi) and arsenic trioxide, of which there is a considerable distance to go before their biomarker exploration. SYKi and BTKi are only used in LN mice currently, and their good symptom-improving effect have a positive impact on their further use in LN patients [72,73,74,75,76]. Furthermore, the phase IIa trial of low-dose intravenous arsenic trioxide in 11 SLE patients showed a decreased level in serum of anti-dsDNA antibody, a potential biomarker, but not C3, C4, IgG, IgA, IgM, anti-Ro/SSA antibody and anticardiolipin antibody [77]. This is a very innovative treatment method that needs further exploration.

## 4. Discussion

We summarize the biomarkers of relevant studies in the order of drug use, from single use, to combinational use, and finally to biological agents (Table 1, Figure 2). Taking the therapeutic mechanism of different drugs as the starting point and the clinical treatment effect as the landing point, we explore whether there are biomarkers that can accurately and effectively indicate the treatment effect. GC and immunosuppressive agents are important drugs to induce remission of LN, and there are many relevant studies. In contrast, biological drugs are relatively few because they are expensive and most of them are still in the early stage of research, and most of the detection indicators are biomarkers traditionally used in clinical practice.

However, as an easily accessible and simple biochemical indicator, biomarkers are of great significance in the understanding and in-depth exploration of diseases. For example, MCP-1 acts as a chemokine whose main role is to activate chemotactic monocyte-macrophages. As a biomarker, MCP-1 is superior to multiple cytokines for LN activity. It is mainly generated through the MAPK signaling pathway (Figure 3). Many studies suggest that MCP-1 could be used as a biomarker reflecting disease activity and prognosis in patients with LN [26,27,78]. Urinary levels of MCP-1 were associated with crescent formation, tubulointerstitial fibrosis and atrophy in patients with LN [79,80]. A prospective cohort study to evaluate the relationship between urine MCP-1 and LN response and long-term renal function showed that persistently elevated urine MCP-1 levels may be associated with unresponsive treatment and continued progression of the disease [80]. In addition, multiple biomarkers combination measurements tend to outperform a single biomarker to better indicate efficacy. A research team used the urine proteome model composed of LPGDS, transferrin, AGP-1, ceruloplasmin, MCP-1 and sVCAM-1 as a potential biomarker group to identify active LN and predict the efficacy of RTX [51,52]. Other research teams have used adiponectin, MCP-1, sVCAM-1 and PF4 as optimal biomarker group models for diagnosing patients with proliferative LN [81]. Furthermore, a study showed that MCP-1 was still stable under various conditions where it had been frozen, freeze-thawed, transported and stored for months in a refrigerator at -80 degrees Celsius [82]. Due to its stable and reliable properties, it may be a biomarker worth considering for predicting the development, treatment and prognosis of LN. 

Most of the biomarkers are not very sensitive and specific, and there is not enough theoretical basis and practical experience for these to be widely used in clinical practice. Because of different clinical trials, there are various degrees of differences in the source of patients, the pathological type of the disease, the length of the disease and the severity f the disease. This makes it difficult to revalidate these tests. In addition, due to the heterogeneity of each individual, the genotype and phenotype are also different, resulting in differences in each patient’s efficacy even if the same drug is used. Therefore, how to achieve better individualized medication and monitor the efficacy of drugs is a problem that needs to be explored and studied in depth. The simultaneous accuracy and reliability of a biomarker is difficult to achieve to meet clinical requirements, and a combination testing scheme may make up the deficiency. In addition, omics technology is a newly developed technology in recent years, such as genomics, metabolomics, transcriptomics, proteomics technology, etc. These omics technologies have the characteristics of high throughput and integrity and have the advantages of simplicity and non-invasiveness, with unique advantages in screening biomarkers and exploring the mechanism of drug action, providing the possibility for understanding the microenvironment, occurrence, development and immunotherapy mechanism of LN. We look forward to a more perfect combination of assays to determine the prognosis of LN patients and monitor disease progression.

## Figures and Tables

**Figure 1 biomolecules-13-01601-f001:**
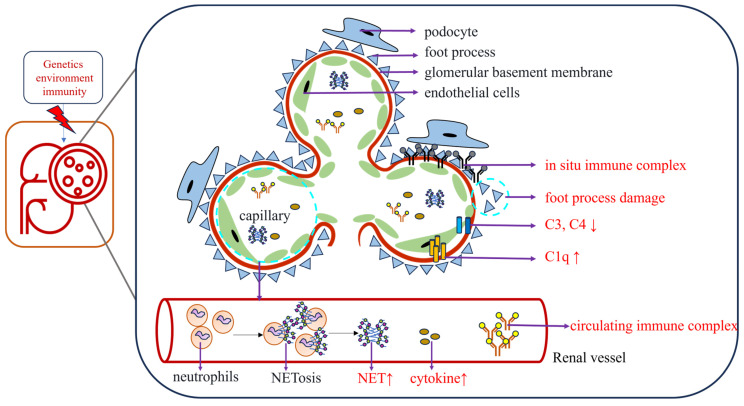
The pathogenesis of LN is related to various factors such as genetics, environmental factors, immunity and NET. Stimulated by internal and external factors, a large amount of abnormal autoantibody produced in the body forms circulating and in situ immune complexes by binding with antigens and deposits in the subepithelial, subendothelial and basement membrane, which changes the levels of various cytokines and complements and mediates kidney damage.

**Figure 2 biomolecules-13-01601-f002:**
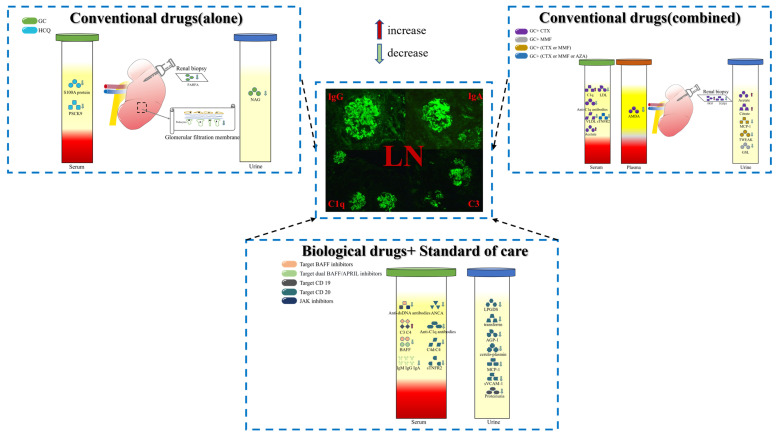
Effective biomarkers in the treatment of LN. Sources of biomarkers are serum, plasma, urine and kidney biopsy. Classification of biomarkers according to traditional drugs and biological drugs. Serum levels of most biomarkers decrease, and a few biomarkers increase (C1q, acetate, citrate, HGF, C3, C4). Different colors are used to distinguish different drugs, and matching drugs and biomarkers use the same colors. The pathological image comes from the Department of Rheumatology and Immunology, the Second Xiangya Hospital of Central South University.

**Figure 3 biomolecules-13-01601-f003:**
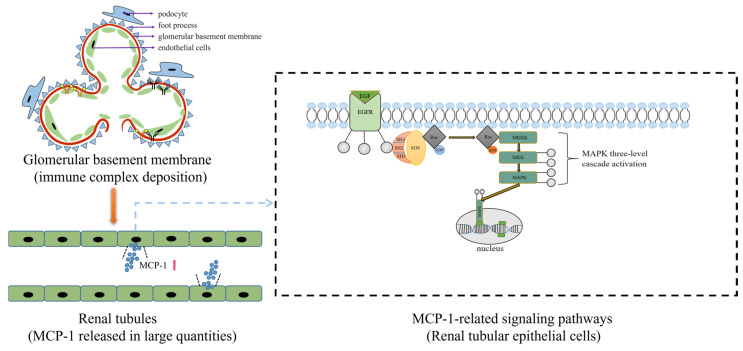
Immune complexes deposit in the glomerulus, stimulating renal tubular epithelial cells. They release a large amount of MCP-1. MCP-1 can activate monocytes and pro-inflammatory immune cells, affect T cell proliferation and immune function and direct leukocyte infiltration. EGF, epidermal growth factor; EGFR, epidermal growth factor receptor; SOS, son of sevenless; GDP, guanosine diphosphate; GTP, guanosine triphosphate; MAPK, mitogen-activated protein kinase; MKKK, MAPK kinase kinase; MKK, MAPK kinase.

**Table 1 biomolecules-13-01601-t001:** Effective biomarkers in the treatment of lupus nephritis.

Drugs	Sample	Biomarkers	Result	Reference
**Conventional drugs**				
GC (alone)	Urine	Podocytes	↓	[6]
	Urine	NAG	↓	[7,8]
	Renal biopsy	FAHFAs	↓	[9]
HCQ (alone)	Serum	S100A8, S100A9	↓	[10]
	Serum	PCSK9	↓	[11]
GC + CTX	Serum	C1q	↑	[13]
	Serum	Anti-C1q antibodies	↓	[14]
	Serum	LDL, VLDL	↓	[17]
	Serum, Urine	Acetate	↑	[17,18]
	Urine	Citrate	↑	[18]
	Renal biopsy	HGF/TGFβ1 ratio	↑	[20]
	Plasma	AMDA	↓	[22]
GC + MMF	Urine	GSL	↓	[24]
GC + (CTX or MMF)	Urine	MCP-1, TWEAK	↓	[32]
GC + (CTX or MMF or AZA)	Serum	sTNFR2	↓	[33]
**Biological drugs + Standard of care (GC, HCQ and immunosuppressive agent)**				
BLM	Serum	Anti-dsDNA antibodies	↓	[37,39]
	Serum	C3, C4	↑	[37,39]
	Serum	BAFF	↓	[38]
Telitacicept	Serum	BAFF, IgG	↓	[43]
ALPN-303	Serum	IgM, IgG, IgA	↓	[44]
CD20	Serum	Anti-C1q antibodies, ANCA	↓	[49]
	Serum	C4d:C4	↓	[50]
	Serum	sTNFR2	↓	[33]
	Urine	LPGDS, transferrin, AGP-1, cerulo-plasmin, MCP-1 and sVCAM-1	↓	[51,52]
CD19	Serum	Anti-dsDNA antibodies	↓	[53]
	Serum	C3, C4	↑	[53]
	Urine	Proteinuria	↓	[53]
IFN inhibitors		Not known yet		
IL inhibitors		Not known yet		
BMS-986165	Serum	Anti-dsDNA antibodies	↓	[71]

GC, glucocorticoid; CTX, cyclophosphamide; MMF, mycophenolate mofetil; HCQ, hydroxychloroquine; AZA, azathioprine; BLM, belimumab; CD20,cluster of differentiation 20; CD19, cluster of differentiation 19;IFN, interferon; NAG, N-Acetyl-β-D-glucosaminidase; FAHFAs, fatty acid esters of hydroxy fatty acids; PCSK9, proprotein convertase subtilisin/kexin type 9; LDL, low-density lipoprotein; VLDL, very-low-density lipoprotein; HGF, hepatocyte growth factor; TGFβ1, transforming growth factor β1; AMDA, asymmetric dimethylarginine; GSL, glycosphingolipid; MCP-1, monocyte chemoattractant protein-1; TWEAK, tumor necrosis factor-like weak inducer of apoptosis; sTNFR2, soluble tumor necrosis factor receptor-2; BAFF, B-cell-activating factor; IgG/M/A, immunoglobulin G/M/A; LPGDS, lipocalin-like prostaglandin D synthase; AGP-1, alpha-1-acid glycoprotein; sVCAM-1, soluble vascular cell adhesion molecule-1.

## Data Availability

Not applicable.

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
