# Peer review of "Biomarkers Associated with Drugs for the Treatment of Lupus Nephritis"

_biomolecules, 2023, doi:10.3390/biom13111601_

Round 1

Reviewer 1 Report

Comments and Suggestions for Authors

The authors introduced interesting concept of specific biomarkers associated with individual drugs in lupus nephritis. Unfortunately, I do not believe we currently really have such biomarkers as all biomarkers used (and usually not validate) are non-specific markers of disease activity or damage.

Generally much data in the manuscript are valuable, but I would suggest the concept of the paper in this way: In the past with non-specific treatment (e.g. GC, cyclophosphamide, mycophenolate, or HCQ) we had no chance to have specific biomarkers. With the introduction of biologic and targeted treatment the situation changed and it may now be the case

Comments:

1.       Introduction is too long and should more concerned to general description of biomarkers of activity and damage in lupus nephritis, how these biomarkers are related to specific damage of different renal cells, etc.

2.       Biomarkers of glucocorticoids alone – the authors mention urinary podocytes, NAG, and FAHFAs, but all these markers are quite non-specific and will be also influenced by any other treatment of LN, so we will not be able to „dissect“ what was done by glucocorticoids and other drug used in treatment. I completely agree with the statement of the authors that glucocorticoids are no longer used in monotherapy in LN

3.       In a similar way ADMA is mentioned as putative marker of response to cyclophosphamide, but it seems to be doubtful as the patient treated with cyclophosphamide were definitely also treated by corticosteroids and ADMA was not tested in patients treated with MMF or biologic treatment.

4.       It is in similar vein with the prognostic role of GSL. It was not studied e.g. in Euro-Lupus, but it seems to be quite non-specific

5.       Autoantibodies are mentione only in chapter on combined treatment. Except for anti-C1q antibodies the authors should also mention at least anti-ds-DNA Ab, anti-nucleosome AB (or antichromatine Ab) and anti-CRP Ab

6.       Also TWEAK and MCP-1 are non-specific markers of apoptosis and inflammation and are increased not only in LN, but MCP-1 is increased also in IgAN or ANCA-associated vasculitis.

7.       The only part of the paper which relates to biomarkers specific to biologic and targeted treatment, e.g.  belimumab and atacicept and similar drugs.  So it means that with targeting specific molecules we may also have specific biomarkers

Comments on the Quality of English Language

Generally the text is readable and understandable, but revision by the native speaker would be valuable

Author Response

Thank you very much for taking the time to review this manuscript entitled “Biomarkers associated with drugs for the treatment of lupus nephritis”(ID: biomolecules-2630071). Please see the attachment for specific responses.

Reviewer 2 Report

Comments and Suggestions for Authors

The manuscript reviews the potential biomarkers related to lupus nephritis therapeutics. The manuscript is interesting, however, in my opinion, it requires major revision as there are some paragraphs describing the use of different agents in the lupus nephritis therapy, but they do not include any specific biomarkers (lines 345-353, 405-412, 427-440) as it is expected based on manuscript title. Moreover, according to authors there are no drugs acting as Janus kinase inhibitors approved for lupus nephritis, therefore, in my opinion the paragraph on Janus kinase inhibitors is not necessary in the manuscript. The manuscript in some parts is focused on the safety and efficacy of the listed drug, therefore, the authors should consider changing the manuscript title or its content for keeping coherence. The other major remark concerns the lack of methodology description. The authors did not specify the time interval of the literature search, mention the literature database searched or provide the terms used during database search.  

Minor remarks are listed below:

·        Are these sentences related to the same study (lines 81-83): “In a study, it was found urinary NAG excretion in the group of patients with LN, rheumatoid arthritis (RA) versus healthy control [7]. In the LN group, NAG excretion was higher than in the RA group and healthy control.” Please, clarify.

·        This part of the text seams missing some information (lines 90-91): “fatty acid esters of hydroxy fatty acids (FAHFAs) are a newly discovered class of biologically active lipid molecules. a team used lipidomics techniques in LN mice model [9].” Please, correct.

·        What kind of metabolites did the authors mean (line 172): “serum metabolite levels”. Were they drug metabolites or other? Please specify.

·        The following sentences do not match the citation given in the references (lines 172-177): “In 2018, Sujata Ganguly et al. analyzed serum metabolite levels for 6 months before and after prednisolone with high-doses or low-doses CTX maintenance therapy in 18 patients with proliferative LN using nuclear magnetic resonance-based metabolomics. After 6 months of treatment, the level of lipid metabolites predominantly LDL and very low-density lipoproteins (VLDL) in the serum decreased significantly, while the level of acetate increased [21]”. The reference No 21 was written by Sundaram TG et al. and published in 2019. Did the authors mean to cite this article or other?

Comments on the Quality of English Language

 The language revision should be performed throughout the manuscript as there are some mistakes such as “the results of patients who were remission by CTX” (line 111), “it plays to drug synergy” (line 150), “mon-itoring” (line 166), “spec-ificity” (line 219), “exper-iment” (line 307), “the did not affect” (line 263). There are also many too long sentences, which are difficult to understand (e.g., line 177-183, lines 210-216) or lack some data: “As a member of the tumor necrosis factor superfamily.” (line 203).

Author Response

(The authors gave the same response as above.)

Round 2

Reviewer 2 Report

Comments and Suggestions for Authors

All my comments have been carefully corrected and answered.